# Performance of Various Filtering Media for the Treatment of Cow Manure from Exercise Pens—A Laboratory Study

Elizabeth Álvarez-Chávez [1,2,*], Stéphane Godbout [2], Alain N. Rousseau [3], Patrick Brassard [2] and Sébastien Fournel [1]

1   Département des sols et de génie agroalimentaire, Université Laval, Québec City, QC G1V 0A6, Canada; sebastien.fournel@fsaa.ulaval.ca
2   Research and Development Institute for the Agri-Environment (IRDA), Québec City, QC G1P 3W8, Canada; stephane.godbout@irda.qc.ca (S.G.); patrick.brassard@irda.qc.ca (P.B.)
3   INRS-ETE/Institut National de la Recherche Scientifique-Eau Terre Environnement, 490 rue de la Couronne, Québec City, QC G1K 9A9, Canada; alain.rousseau@inrs.ca
\*   Correspondence: paz-elizabeth.alvarez-chavez.1@ulaval.ca

**Abstract:** During summer and winter months, pastures and outdoor pens represent the conventional means of providing exercise for dairy cows housed in tie-stall barns in the province of Québec, Canada. Unfortunately, outdoor pens require large spaces, and their leachates do not meet Québec's environmental regulations. Therefore, there is a need to develop alternative approaches for these so-called wintering pens. A sustainable year-long approach could be a stand-off pad consisting of a filtering media to manage adequately water exiting the pad. Different filtering materials can be used and mixed (gravel, woodchips, biochar, sphagnum peat moss, sand, etc.). To find the best material and/or mixes, a laboratory study was carried out using 15 PVC pipes (5 cm in diameter and 50 cm long) to test five different combinations of materials over a 3-week period. Different contaminant-removal efficiencies were achieved with the alternative materials, including for chemical oxygen demand (11–38%), phosphates (8–23%), suspended solids (33–57%), and turbidity (23–58%). Alternative treatments with sand, sphagnum peat moss, and biochar improved the filtration capacity when compared to the conventional material (woodchips). However, after three weeks of experimentation, the treatment efficiency of sand gradually decreased for pollutants such as suspended solids and phosphates.

**Keywords:** dairy production; stand-off pad; water treatment; urine filtration; sphagnum peat moss; biochar; woodchips; nitrogen removal; Québec

## 1. Introduction

Dairy production in Québec, Canada, is characterized by family-run farms (70 cows/barn on average) [1] using tie-stall housing in a large proportion (91%) [2]. This housing system is controversial because it restricts the voluntary movement of the cows and their ensuing social behavior [3,4]. Moreover, tie-stalls are associated with problems including hock swellings and abrasions, neck lesions, broken tails, and lameness [5–7]. Consequently, for lactating cows any exercising opportunities as often as possible have become mandatory and an integral part of the Dairy Cattle Code of Practice of the National Farm Animal Care Council [8].

Labelle [9] provides a list of practical solutions for dairy cattle. Accordingly, producers have the possibility to: (1) renovate and convert their barns to free-stall housing; (2) build new ones; or (3) simply add an exercise yard to their existing tie-stall barns. The first two options are quite expensive (more than CAD 300,000). On the other end, indoor exercise pens can be a suitable option [10] but raise concerns that reduced space allowances might increase aggressiveness within the herd, restrict natural behavior, and enhance abiotic environmental sources of stress and confinement-specific stressors [11,12]. Further, when given the choice between indoor and outdoor areas, most cows prefer the latter [10]. This

study also noted that cows, when outdoors, performed a range of normal activities such as lying down and feeding even during the harsh winter conditions of eastern Canada. Since pastures are not functional during winter (low temperatures and presence of a snowpack), a wintering pen and a vegetative filter strip have become the traditional method to provide exercise opportunities. However, this system must meet Québec's environmental regulations stipulating that contaminated runoff water from livestock yards must not percolate into ground waters or discharge into surface waters. Unfortunately, after 15 years of research on wintering sites for beef cattle, it has been demonstrated that none of the proposed configurations is continuously 100% effective year after year [13].

Consequently, a similar outdoor alternative effective all year long could consist of a stand-off pad (SOP) made of a filtering material overlying an impermeable lining with drainage pipes discharging into a manure tank [14]. This system is currently well-established in Ireland, the United Kingdom, and New Zealand, where it is considered as an economic alternative of wintering dairy cows without compromising animal performance, health, or welfare while also treating soiled water through a matrix, which acts as a filtration and retention media of slurry solids while capturing all the remaining effluent. The possible reduction in gaseous emissions of ammonia and nitrous oxide represents another advantage of SOPs since the rapid drainage of effluent from the surface would reduce the scope for rapid urea hydrolysis and subsequent nitrogen diffusion into the air [14–16]. Despite their advantages, the SOPs ultimately generate a leachate that can still contain nutrients and fecal microbes. To improve the quality and limit the quantity of effluent perhaps to the point where the drainage and liner systems are no longer required (reduction of construction costs), the choice of the filtering media has become critical [17]. The most relevant characteristics for a good filtering material are: high specific surface area, high void fraction, large free-passage diameter, resistance to clogging, inert material of construction, cost effectiveness, good mechanical strength, light weight, flexibility in overall shape, wettability for better biofilm growth, light attenuation of nitrifying bacteria, and facility of maintenance [18].

Conventional surface material options for SOPs include rock products (gravel, stone, and soft rock), wood products (woodchips, post-peelings, sawdust, and bark), and sand. Rock products are easy to source in some areas, but they are not recommended for lying provision [16]. Wood products have been used for wastewater treatment because they are a source of carbon for microbial respiration and denitrification, which benefits nitrogen (N) removal from wastewater under anaerobic conditions [19]. According to Jackson [20], woodchips (10–20 mm in size) can absorb almost three times their weight, and thus, up to 90% removal of N and phosphorus (P) from livestock manure deposition can be achieved. Meanwhile, slow sand filters can efficiently remove various waterborne pathogens, including viruses, bacteria, and protozoan cysts of giardia and cryptosporidium enteroparasites. However, efficiency is site-specific depending on operating parameters, such as temperature, filtration rate, particle size of medium, and bed depth [21]. In addition, sand is not as comfortable as wood products for dairy cows [22].

An alternative filtering material may be biochar. Some studies have shown that biochar can improve nitrate removal by increasing storage volume and residence time thanks to high surface charge density, high specific surface area, and high micropore volume, making it an effective sorbent [23]. Biochar can also simultaneously remove different types of contaminants, including metals/metalloids and microbial and organic contaminants. Removal of contaminants by biochar could vary based on several factors: contaminant characteristics, biochar properties, and treatment conditions [24]. In addition, this material has been used for plant biomass enhancement when mixed with compost to prevent leaching of N, P, and organic carbon [25].

Sphagnum peat moss, a light-brown to black organic material with a large specific surface area, has a very porous structure (95%) and can be used as a low-cost adsorbent with a high treatment capacity of aqueous solutions. It has also been investigated for the

removal of heavy metals in wastewater [26] and possesses excellent adsorption properties for organic and inorganic molecules, including soluble organic micro pollutants [27,28].

Limited information is available on the impact of different organic filtering materials to reduce nutrient loading in SOPs [19,29–32]. Furthermore, there is a lack of information on the treatment performances of various filtering material combinations of manure from SOPs. Therefore, the objective of this study was to investigate the performances of different material combinations with different adsorption properties (alternative treatment) in comparison to a conventional material, that is, the use of woodchips to treat dairy manure. This investigation was carried out at a laboratory scale as part of a study focusing on the development of an alternative exercise pen for dairy cows housed in tie-stall barns under northern climate conditions.

## 2. Materials and Methods

### 2.1. Experimental Setup and Design

A fifteen-PVC-column (5 cm in diameter and 50-cm long; Figure 1) set up housed at the Research and Development Institute for the Agri-Environment (IRDA) facility in Québec City (QC, Canada) was used to treat synthetic dairy cow manure during 3 weeks between 17 August and 4 September 2020. Synthetic manure was used in order to have a known and consistent concentration at laboratory scale. At the base of the filters, a 0.2 mm mesh was installed to avoid any material loss. To collect leachate samples, the bottom of the columns was connected to a flexible tube, linked to a 1000 mL container.

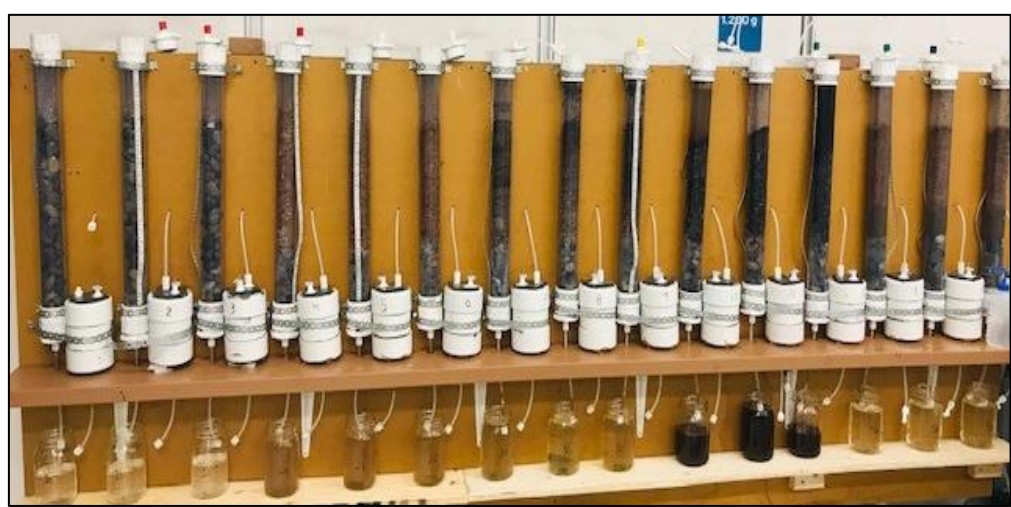

**Figure 1.** PVC columns in IRDA's laboratory.

The columns, kept at ambient temperature (~20 °C), were divided into five combinations (three replicates) of filtering materials. The first treatment (500 mm of gravel) was used as a control (referred to as CT; Table 1) [30], while the other treatments had an absorbent layer (300 mm) and a subsurface layer made of gravel (200 mm), the latter to mimic subsurface drainage system of a SOP. The layering respected the proportions of a commercial SOP according to DairyNZ [16]. The gravel was acquired from a local supplier (S. Boudrias Inc., Laval, QC, Canada), while black spruce woodchips for the conventional wood treatment (CW; Pépinière St-Modeste, QC, Canada) and sphagnum peat moss (Miracle-Gro, Marysville, OH, USA) were from well-known suppliers. The latter was washed under running water to remove impurities and sieved to obtain fractions larger than 2 mm before being mixed with woodchips for the treatment referred to as sphagnum alternative (MA). The biochar, which was produced by pyrolysis of forest residues in a vertical auger reactor (with following characteristics: 559 °C temperature, 61 s of residence time and 3 L/min of nitrogen flow), was mixed with woodchips for the biochar alternative treatment (BA). Sand (Techniseal, Montréal, QC, Canada) was washed and sieved to obtain 1.18 mm fraction

before being incorporated into columns of the sand alternative treatment (SA). All products are shown in Figure 2. After the columns were filled, 2 L of demineralized water was added to each one to wash out the filtering media.

**Table 1.** Properties of the filter media uses in this study.

| Treatment | Composition | | | | |
|---|---|---|---|---|---|
| | Section | Material | Depth (mm) | Grading (mm) | Quantity (g) |
| Control (CT) | Top | Gravel | 500 | 25–50 | 1034 |
| Conventional wood (CW) | Top | Woodchips | 300 | 10–20 | 190 |
| | Bottom | Gravel | 200 | 25 | 580 |
| Biochar alternative (BA) | Top | Woodchips/biochar (20% *v/v*) | 300 | 10–20/<5 | 115/16.9 |
| | Bottom | Gravel | 200 | 25–50 | 580 |
| Sphagnum alternative (MA) | Top | Woodchips/sphagnum peat moss (80% *v/v*) | 300 | 10–20/0–20 | 28/230 |
| | Bottom | Gravel | 200 | 25–50 | 580 |
| Sand alternative (SA) | Top | Woodchips | 200 | 10–20 | 94 |
| | Middle | Sand | 100 | 0.25–1.2 | 290 |
| | Bottom | Gravel | 200 | 25–50 | 580 |

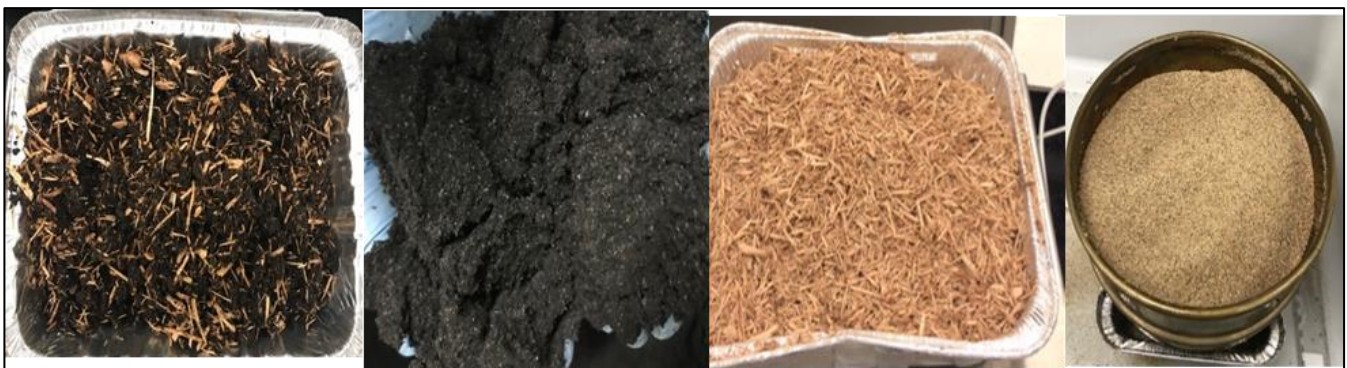

**Figure 2.** Filtering materials (from (**left**) to (**right**)): woodchips, biochar, sphagnum peat moss, and sand.

## 2.2. Application of Synthetic Dairy Cow Manure and Rainwater

Synthetic manure had the chemical composition of water contaminated by livestock feces and urine (similar to runoff from a SOP) [19]. Livestock-soiled water, that is dairy cow manure, is similar in composition to cattle slurry, but it is more diluted, which is of interest for complete filtration tests.

The components and the amounts needed to prepare a 14 L solution are listed in Table 2. All reagents were of analytical grade and used without further purification. Demineralized water was used to avoid the high chlorine content of tap water. The synthetic manure was prepared each week and stored in a cool room. Before being manually poured on the columns, it was mixed to ensure a homogeneous mixture. Based on a hydraulic loading rate of 30 L m$^{-2}$ d$^{-1}$ [19,33], 0.268 L of synthetic manure was applied 5 days per week on each column.

**Table 2.** Composition of synthetic dairy cow manure diluted with 85 L of water (values converted from Healy [34]).

| Components | Amounts (g) |
|---|---|
| Glucose | 87 |
| Yeast | 13 |
| Dried milk | 8.6 |
| Urea | 10 |
| $NH_4Cl$ | 21 |
| $Na_2PO_4\ 12H_2O$ | 33 |
| $KHCO_3$ | 22 |
| $NaHCO_3$ | 57 |
| $MgSO_4 \cdot 7H_2O$ | 22 |
| $FeSO_4 \cdot 7H_2O$ | 0.9 |
| $MnSO_4 \cdot H_2O$ | 0.9 |
| $CaCl \cdot_6 H_2O$ | 0.2 |
| Bentonite | 18 |

To simulate recurrent rainfall events on a SOP, 80 mL of demineralized water was manually added to each filtering medium 2 days per week after the addition of synthetic dairy cow manure (i.e., wastewater). This value was calculated according to Environment and Climate Change Canada weather statistics considering the average rainfall during summer rainy days in Deschambault for the years 2014–2019 (8 mm), as recorded in the Québec City region (Climate ID 7011982; Canada, 2020), multiplied by the area of the PVC tube (0.0019 m$^2$) and the number of days synthetic manure was added.

*2.3. Physicochemical Analyses of Effluents*

A 100 mL sample of effluent was collected three times per week according to the sampling guides for environmental analysis (Ministère du Développement durable, de l'Environnement et de la Lutte aux changements climatiques, 2012). The samples were kept at 4 °C in a laboratory refrigerator after pH and electrical conductivity were determined using a multi-parameter meter (HI991301, Hanna Instruments, Lingolsheim, France). Turbidity (Tb), suspended solids (SS), chemical oxygen demand (COD), total nitrogen (TN), nitrates ($NO_3$-N), and phosphates ($PO_4$-P) were determined weekly using a colorimeter (DR 900, HACH, Loveland, CO, USA), reagents, and deionized water (Table 3). The removal efficiencies of each column (*R*; %) were calculated according to Equation (1) [27].

$$R = \frac{(Ci - Ce)}{Ci} \times 100 \tag{1}$$

where *Ci* and *Ce* are the initial and equilibrium concentrations (mg L$^{-1}$), respectively.

*2.4. Statistical Analyses*

A comparative statistical analysis using SAS 9.4 software (version 9.4, SAS Institute Inc., Cary, NC, USA) was performed to determine whether the averages were significantly different between treatments. Due to the small number of samples and a probable slight deviation from normality in the effluent samples, the Tukey–Kramer test was used to mitigate the risk of Type 1 error when comparing several treatments. A significant difference was defined at a probability level of 0.05.

**Table 3.** Laboratory methods used for analysis.

| Equipment | Parameter | Method | Québec's Environmental Regulation [35,36] |
|---|---|---|---|
| Multi-parameter Hanna (model: HAHI991301) | pH, Conductivity | Electrometric method | 6.5–8.5, <1500 μS/cm |
| DR900 Colorimeter HACH | Turbidity (Tb; 21 to 1000 FAU) | Absorptometric method 8237 HACH | 5 (NTU) |
| | Suspended solids (SS; mg L$^{-1}$) | Photometric method 8006 HACH (5–750 mg L$^{-1}$) | 15 mg L$^{-1}$ |
| | Chemical oxygen demand (COD), low (3–150 mg L$^{-1}$) and high (20–1500 mg L$^{-1}$) range | Colorimetric method 8000 HACH: method 410.4 EPA | |
| | Total nitrogen (NT; 2–150 mg L$^{-1}$) | Persulfate digestion method, method 10072 HACH | |
| | Nitrates (NO$_3$-N; 0.2 to 30.0 mg L$^{-1}$) | Chromotropic acid method 10020 HACH | |
| | Phosphates (PO$_4$-P; 0.3 to 45.0 mg L$^{-1}$) | Molybdovanadate method 8114: method 4500-P-E | 1.0 mg L$^{-1}$ |

## 3. Results and Discussion

### 3.1. Synthetic Dairy Cow Manure

Table 4 introduces the physical and chemical properties of the synthetic dairy cow manure used for this study. As shown, the concentration of PO$_4$-P was significantly higher than those reported in the literature [31,34], while it was the opposite for suspended solids.

**Table 4.** Comparison of the physical and chemical properties of the synthetic manure with those reported in the literature.

| Parameter | Synthetic Manure | Healy, Rodgers [34] | Murnane [31] |
|---|---|---|---|
| pH | 7.7 | 7.9 | 7.22 ± 0.71 |
| Turbidity (Tb; FAU) | 79 ± 26 | - | - |
| Suspended solids (SS; mg L$^{-1}$) | 78 ± 21 | 457 | 874 ± 614 |
| Chemical oxygen demand (COD; mg L$^{-1}$) | 988 ± 104 | 1395 | 2798 ± 1503 |
| Total nitrogen (TN; mg L$^{-1}$) | 96 ± 23 | 66 | 81.5 ± 34.1 |
| Nitrates (NO$_3$-N; mg L$^{-1}$) | 2.3 ± 2 | 12.5 | |
| Phosphates (PO$_4$-P; mg L$^{-1}$) | 50.4 ± 32 | 6.8 | 29.8 ± 14.4 |

### 3.2. Effluent pH

Average effluent pH from treatments CT, CW, BA, MA, and SA was as follows: 6.7 ± 0.6, 6.4 ± 0.4, 6.9 ± 0.3, 6.3 ± 0.3, and 6.6 ± 0.4. Treatment MA was slightly more acidic since the pH of sphagnum peat moss has a value around 4.0 [37]. pH has been reported to have an important role in the degradation of pollutants, as it can affect the structure and properties of materials as well as the biological activity. Usually, the desirable pH range is 7.0–8.0, so it was not a limiting factor in the results for all the proposed treatments [38].

### 3.3. Weekly Effluent Concentrations

Figure 3 shows the evolution of contaminant concentrations over the 3-week experiment for each treatment. Total SS tended to increase week after week for each treatment, demonstrating an accumulation of solid matter in the biofilters. The higher CT results (52–94 mg L$^{-1}$) suggest that other treatments (17–68 mg L$^{-1}$) had better removal efficiencies.

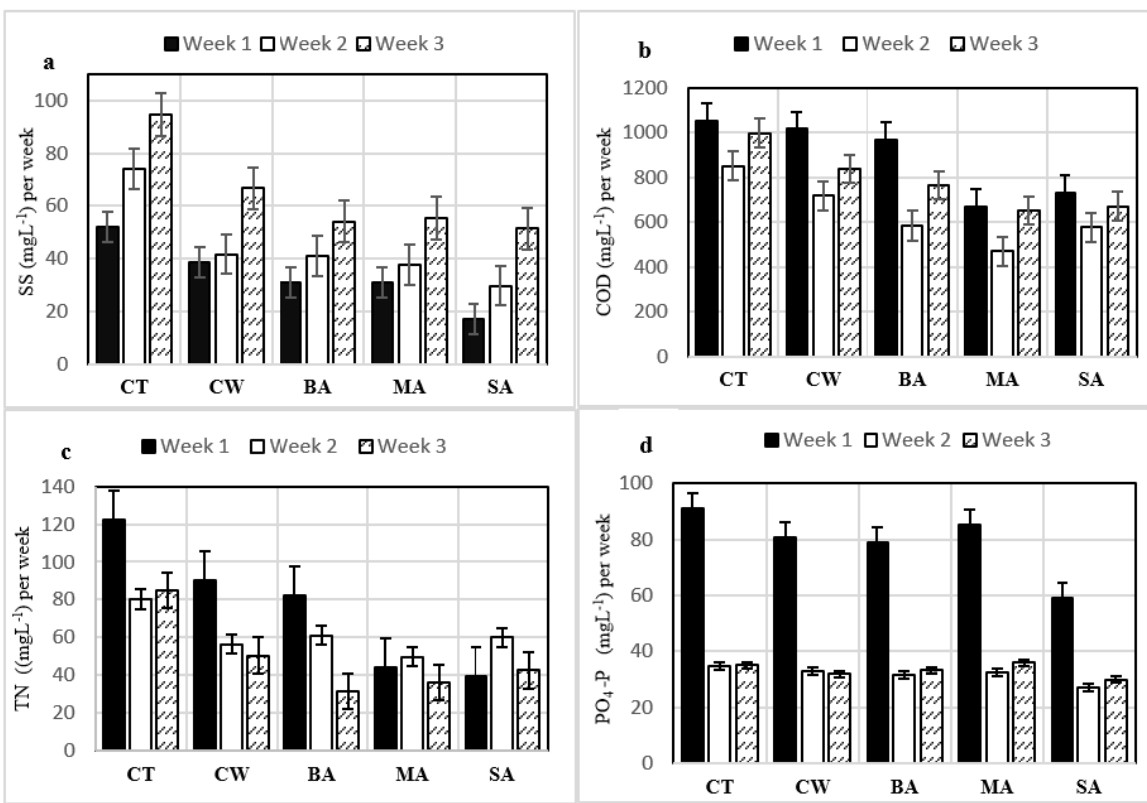

**Figure 3.** Concentrations per week (1–3) of SS (**a**), COD (**b**), TN (**c**), and PO4-P (**d**) for treatments CW (woodchips), BA (woodchips + biochar), MA (woodchips + sphagnum peat moss), and SA (woodchips + sand).

For all treatments, the COD reached a maximum level during the first week, then dropped for the second week before increasing in the last week. This is due to the COD concentration of the water used for the synthetic manure, which was slightly higher during the first week. The MA treatment obtained the lowest COD value (470 mg $L^{-1}$), which is in line with Kõiv [39] who demonstrated that sphagnum peat moss is a highly efficient filtering material for removing organic matter.

Total nitrogen concentration generally decreased over time for treatments CT, CW, and BA, while it stayed relatively stable for treatments MA and SA. In fact, the concentration during the second week slightly increased before decreasing close to the initial level during the third week. This may be due to a period of adaptation during the first days. In addition, certain materials were clearly more compatible for TN removal than others. For instance, the BA treatment gradually decreased the concentration from 82.0 to 31.3 mg $L^{-1}$. Other studies have clearly shown the potential of biochar in the removal of organic and inorganic compounds [40,41].

The concentration of PO$_4$-P decreased over time for all treatments, suggesting that the filtration process is better at the beginning, as the materials are not saturated with organic matter. The minimum concentration was obtained with the SA treatment during the second week (27 mg $L^{-1}$), and by the third week, it began to increase again (29 mg $L^{-1}$), indicating that the filter became saturated. The above results coincide with the study of Achak, Mandi [42], where it was found that the flow rate in a sand filter decreases with respect to time, so the technical and economic feasibility of using the SA treatment for cow manure still needs to be established. Table 5 shows the weekly removal efficiency for Tb, SS, COD, TN, and PO$_4$-P for all treatments (CW, BA, MA, and SA).

**Table 5.** Weekly and average (Avg) removal efficiencies [1] for turbidity (Tb), suspended solids (SS), chemical oxygen demand (COD), total nitrogen (TN), and phosphates (PO₄-P) for each treatment with respect to the control.

| Treatment | Week | Parameter | | | | | |
|---|---|---|---|---|---|---|---|
| | | Tb | SS | COD | TN | $PO_4$-P | $NO_3$-N |
| CW | 1 | 25.1 | 26.1 | 3.49 | 26.4 | 11.2 | 73.6 |
| | 2 | 29.6 | 43.5 | 15.6 | 29.8 | 5.61 | 68.8 |
| | 3 | 14.6 | 29.6 | 15.9 | 40.7 | 8.67 | 3.4 |
| | Avg | 23.1 | 33.1 | 11.7 | 32.3 | 8.51 | 48.6 |
| BA | 1 | 25.4 | 40.9 | 8.11 | 32.9 | 13.5 | 90.5 |
| | 2 | 31.4 | 44.7 | 31.4 | 24.0 | 4.18 | 13.1 |
| | 3 | 36.4 | 42.8 | 23.5 | 63.1 | 4.95 | 37.9 |
| | Avg | 31.1 | 42.8 | 21.0 | 40.0 | 7.56 | 47.2 |
| MA | 1 | 11.5 | 40.4 | 36.4 | 64.0 | 6.36 | 0.0 |
| | 2 | 25.0 | 49.2 | 44.7 | 38.1 | (3.5) | 84.0 |
| | 3 | 31.0 | 41.5 | 34.7 | 57.6 | (2.5) | 0.0 |
| | Avg | 22.5 | 43.8 | 38.6 | 53.3 | 0.07 | 28.0 |
| SA | 1 | 65.0 | 67.1 | 30.3 | 67.5 | 34.9 | 13.7 |
| | 2 | 55.4 | 59.7 | 32.2 | 25.3 | 17.2 | 84.4 |
| | 3 | 54.7 | 45.7 | 32.7 | 50.0 | 14.3 | 91.5 |
| | Avg | 58.4 | 57.5 | 31.7 | 47.6 | 22.2 | 63.0 |

[1] Numbers in parentheses represent negative removals or increases.

### 3.4. Contaminant Removal Efficiencies

3.4.1. Suspended Solids (SS) and Turbidity (Tb)

Figure 4 shows the results of the pairwise contrasts for the three main parameters (SS, COD, and TN) for each treatment (CW, BA, MA, and SA) with respect to the control (CT). As introduced in Table 5, the average SS removal efficiencies for CW, BA, MA, and SA treatments were as follows: 33.1%, 42.8%, 43.7%, and 57.5%, respectively. Adding alternative materials to the conventional treatment can improve the removal of pollutants due to smaller particle size, leading to longer hydraulic retention time [41,43]. For example, the hydraulic conductivity of sand is 40 times lower than woodchips [31]. However, this characteristic led to the plugging of the sand filter at the end of week 3, which explains a decrease of SS removal efficiency from week 1 to 3 (Table 5: 67.1%, 59.7%, and 45.7%, respectively). Furthermore, the average removal for Tb for treatments CW, BA, MA, and SA were as follows: 23.1%, 31.1%, 22.5%, and 58.4%. The SA treatment achieved the best removal ($p < 0.05$), which is in agreement with the result obtained for SS removal. This corroborates the rule of thumb that the removal of Tb is a simple and fast estimation of SS [41].

3.4.2. Removal of Organic Matter (COD)

As shown in Figure 4, the COD removal efficiencies for CW, BA, MA, and SA treatments were as follows: 11.7%, 21.0%, 38.6%, and 31.7%, respectively. The MA and SA treatments outperformed ($p < 0.05$) the conventional treatment. The study by Murnane [31] reported that sand was superior to woodchips for COD removal, which is linked to its high hydraulic retention time. However, these authors concluded that the use of sand on a large scale would raise costs. The good performance for MA can be attributed to the sphagnum peat moss having a porous structure and polar functional groups, which allow this material to be an efficient adsorbent for dissolved contaminants such as metals and organics [44,45].

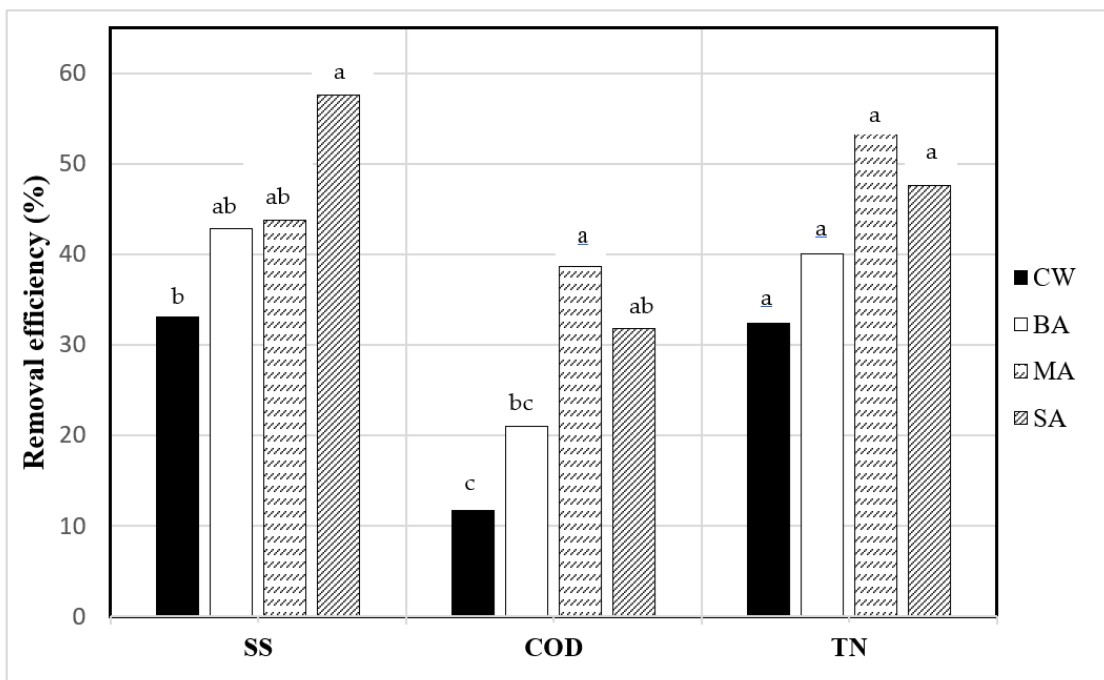

**Figure 4.** Average removal efficiencies for suspended solids (SS), chemical oxygen demand (COD), and total nitrogen (TN) for each treatment relatively to the control. Bar values followed by the same letter for the same parameter are not significantly different at $p = 0.05$ as determined by pairwise contrasts.

### 3.4.3. Total Nitrogen (TN) and Nitrates ($NO_3$-N)

As shown in Figure 4, the TN-removal efficiencies during the experiment for treatments CW, BA, MA, and SA were as follows: 32.3%, 40.0%, 53.2%, and 48.0%, respectively. The three alternative treatments obtained better TN removals than the CW treatment ($p > 0.05$). The addition of biochar, sphagnum peat moss, and sand could have increased the denitrification capacity of woodchips by: (1) changing the hydraulic properties of the filter which increased the contact time of contaminated water or (2) changing the chemistry of pore water [43,46].

The percentage removal rates for $NO_3$-N for the CW, BA, MA, and SA treatments were as follows: 48.6%, 47.2%, 42.2%, and 63.0%, respectively. No treatment was significantly different among them ($p > 0.05$). However, there was a numerical difference for treatment SA, while the results of treatments CW, BA, and MA were very similar.

Therefore, although significant nitrate removal was achieved in the 500 mm deep filters, especially with the SA treatment, which obtained a numerical but not significant difference ($p > 0.05$), the removal process was by physical filtration of SS rather than by biological transformations [31]. Although some studies recommended using alternative materials such as biochar and sand to increase nitrate reduction, the present results were not favorable for nitrogen removal in general [15,43,47]. This might be due to the absence of inoculation, which could have favored biological activity.

### 3.4.4. Phosphates ($PO_4$-P)

As introduced in Table 5, the average percent removal during the 3 weeks of experimentation for the CW, BA, MA, and SA treatments was as follows: 8.5%, 9.3%, 3.3%, and 22.1%, respectively. The results suggest that the SA biofilter was significantly the best at retaining phosphates ($p < 0.05$), which are associated with the removal efficiency of solids according to the study carried out by Murnane [31].

An additional explanation for phosphate removal in sand substrates is that P is bound to the medium mainly because of adsorption and precipitation reactions with calcium

(Ca), aluminum (Al), and iron (Fe). However, the efficiency of P removal is usually high at the beginning and then decreases with time [48] as the adsorption capacity of the sand is reduced. This is due to the saturation of the medium, which causes a decrease in flow rate with time. This was noticed during the experiment since the SA treatment started to become clogged after the second week.

Since conventional filtration systems are not efficient in retaining phosphates, biological treatments are always considered as an alternative to improve phosphate removals even with materials such as sand [34,49]. Biochar and sphagnum peat moss, despite being reported as absorbent materials capable of retaining P [23,48], were not a factor in the BA/MA treatments since their phosphate removal during this experiment was almost null.

## 4. Conclusions

This study had for its objective to investigate the performance of conventional and alternative filtering materials to treat dairy manure at laboratory scale. The experimental design consisted of 15 experimental filters using five different materials and mixes for a total of three repetitions per treatment. The main conclusions of a three-week experiment using synthetic cow manure are that the best treatment for the removal of $PO_4$-P, SS, and Tb was the SA treatment, with removal efficiencies of 22%, 58%, and 58%, respectively. However, during the third week, the filtering media began to clog, resulting in difficult filtration. This could represent additional maintenance costs for producers, and thus, this treatment is not recommended for use on dairy farms. Following the SA treatment, the MA treatment had higher removal efficiencies for organic matter and TN, with values of 38% and 53%, respectively, while the CW treatment achieved removal efficiencies of 11% and 32%, respectively. Nitrate removal, although not significantly different ($p > 0.05$), was higher with the BA treatment, with an average removal efficiency of 47%. Consequently, future work should carry out large-scale tests using filtering media made of a mixture of biochar and sphagnum peat moss to take advantage of their individual effects.

**Author Contributions:** Conceptualization, S.F. and S.G.; methodology, E.Á.-C. and S.F.; validation, S.F and S.G., formal analysis, E.Á.-C.; investigation, E.Á.-C.; resources, S.G.; writing—original draft preparation, E.Á.-C.; writing—review and editing, S.F., A.N.R., S.G., and P.B.; visualization, E.Á.-C. and S.F.; supervision, S.F.; project administration, S.F.; funding acquisition, S.G. All authors have read and agreed to the published version of the manuscript.

**Funding:** This project was funded by the 2018–2023 Innov'Action agroalimentaire (IA119559) and the 2018–2023 Partenariat pour l'innovation en agroalimentaire (PPIA04) programs under the Canadian Agricultural Partnership between the Ministère de l'Agriculture, des Pêcheries et de l'Alimentation du Québec (MAPAQ) and the federal government. We also gratefully acknowledge the financial and technical support of Université Laval and the Research and Development Institute for the Agri-Environment (IRDA).

**Institutional Review Board Statement:** Not applicable.

**Informed Consent Statement:** Not applicable.

**Data Availability Statement:** Not applicable.

**Acknowledgments:** The authors would especially like to thank Joahnn Palacios (IRDA) for his technical assistance.

**Conflicts of Interest:** The authors declare no conflict of interest.

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
