# Peer review of "Performance of Various Filtering Media for the Treatment of Cow Manure from Exercise Pens—A Laboratory Study"

_water, doi:10.3390/w14121912_

Round 1

Reviewer 1 Report

In this manuscript, different filtering materials were used or mixed to treat the dairy manure at a laboratory scale. The contaminant retention capacities of five filters were evaluated and compared to each other. This work should be of interest to the readership of Water. The authors may wish to consider the following comments in a revised version:

  1. Since the harsh conditions in winter, the influence of temperature on the contaminant retention capacities of the filters is suggested to be considered.

  1. The weekly retention ratios of the contaminant for each filter should be included to show the variation of the contaminant retention capacities.

  1. The variations of the water flux or permeate rate of the filters are suggested to be included to show the treatment efficiency.

  1. The introduction part is suggested to be well-organized to show the novelty of this work.

  1. In page 11, line 428-431, “The experimental design consisted in 12 experimental filters using four different materials and mixes for a total of three repetition per treatment.” Actually, in this study, 15 experimental filters using five different materials and mixes were tested.

  1. The most recommended filter should be clearly pointed out in the Conclusions.

Author Response

Reviewer 1

Since the harsh conditions in winter, the influence of temperature on the contaminant retention capacities of the filters is suggested to be considered.

This first work is considered part of a bigger project. In the second phase, one of the variables considered is temperature since we tested similar columns in typical summer and winter conditions. In the first phase, we only planned to test the possible materials to be used in the second phase.

The weekly retention ratios of the contaminant for each filter should be included to show the variation of the contaminant retention capacities.

Table 5 has been modified to include weekly retentions ratios of the contaminant for each treatment.

The variations of the water flux or permeate rate of the filters are suggested to be included to show the treatment efficiency.

Since this experiment was developed at the laboratory level, the variation of water flow was not measured in this first phase since the objective was to test the contaminant retention capacity of the combinations of organic materials, subsequently the second phase of the project is to test the best combination and evaluate its retention capacity with fresh manure by measuring its retention capacity and leachate flux.

The introduction part is suggested to be well-organized to show the novelty of this work.

The introduction was rearranged and we tried to put emphasis on the novelty according to this referee´s comments (See page 3, line 111-119).

In page 11, line 428-431, “The experimental design consisted in 12 experimental filters using four different materials and mixes for a total of three repetition per treatment.” Actually, in this study, 15 experimental filters using five different materials and mixes were tested. Justificacion

In page 11, line 467-468, was changed to “The experimental design consisted in 15 experimental filters using four different materials and mixes for a total of three repetition per treatment“.

Reviewer 2 Report

This manuscript is generally well conceived and written. However, before being accepted for publication the following aspects should be revised:

1. This study novelty, in comparison with international relevant literature are  missing. I have observed that the references used in the Introductory part are not covering the latest research published in the last 10 years, that would facilitate the formulation of the novelty section.

2. Some sentences should be modified in the abstract and main text. For example: "retention results" or "percentage removal rates" should be replaced by removal efficiency (the frequently used term in the scientific literature); removal of "chemical oxygen demand" should be replaced by removal of organic matter since the COD is a measure of the organic material in the wastewater samples.

3. A list with the most frequently used abbreviations should be provided, even if these are explained on their first use. This would also avoid repetitions of abbreviations in the main text.

4. As I know, the COD is measured by the spectrophotometric or colorimetric method. Why do the authors use the term "Absorptometric"?

5. Table 3 should be completed with the maximum allowed concentrations of each parameter in wastewater according to the national legislation. 

6. Did the authors made any comparison (for the composition) between synthetic manure and real manure ?

7. Chemical formulae should be corrected for all the text and tables. They are inconsistent for the superscripts/subscripts and charges. The same stands for the parameters PO4-P and the nitrogen species which are named in different ways.

8. The Conclusions should be written without bullet points.

Author Response

Reviewer 2

The most recommended filter should be clearly pointed out in the Conclusions.

The conclusion was reformulated to pointed out the most recommended filter.

This study novelty, in comparison with international relevant literature are  missing. I have observed that the references used in the Introductory part are not covering the latest research published in the last 10 years, that would facilitate the formulation of the novelty section.

References published in the last 10 years were added (see 4-7,18,26,27,29-32,38,39).

Some sentences should be modified in the abstract and main text. For example: "retention results" or "percentage removal rates" should be replaced by removal efficiency (the frequently used term in the scientific literature); removal of "chemical oxygen demand" should be replaced by removal of organic matter since the COD is a measure of the organic material in the wastewater samples.

Sentences were modified, see lines 22, 411 and 418.

A list with the most frequently used abbreviations should be provided, even if these are explained on their first use. This would also avoid repetitions of abbreviations in the main text.

The guide for authors do not indicate where to add a table of abbreviations. We let the editor decide where to put it. The table of abbreviations is as follows:

Al: aluminium

BA : Biochar alternative

Ca: calcium

COD: Chemical oxygen demand

CT: Control

CW: Conventional wood

FAU : Formazine attenuation unit

Fe: iron

IRDA: Research and Development Institute for the Agri-Environment

MA: Sphagnum alternative

NTU : Nephelometric Turbidity unit

NO3-N: Nitrates

PO4-P: Phosphate

PVC: Polyvinyl chloride

SA: Sand alternative

SOP: Stand-off pad

SS: Suspended solids

Tb: Turbidity

TN: Total nitrogen

As I know, the COD is measured by the spectrophotometric or colorimetric method. Why do the authors use the term "Absorptometric"?

COD test term was changed to colorimetric method, see table 3, page 6.

Table 3 should be completed with the maximum allowed concentrations of each parameter in wastewater according to the national legislation.

Table 3 was completed with the provincial legislation.

Did the authors made any comparison (for the composition) between synthetic manure and real manure ?

This experiment was carried out at laboratory level, synthetic water was used to simulate dairy soiled water since the manure concentration is significantly more concentrated and to compare the concentrations at laboratory level involves other difficulties. Table 4 was supplemented with another concentration of dairy soiled water from another author.

Chemical formulae should be corrected for all the text and tables. They are inconsistent for the uperscripts/subscripts and charges. The same stands for the parameters PO4-P and the nitrogen species which are named in different ways.

The chemical formulas were changed, and tables were corrected.

Conclusions should be written without bullet points

The conclusions were rewritten.

Round 2

Reviewer 2 Report

The authors revised their MS considering all the comments that I have addressed in my review report, therefore this article may be accepted in its current form for publication.